# Additional Value of [^18^F]FDG PET or PET/CT for Response Assessment of Patients with Gastrointestinal Stromal Tumor Undergoing Molecular Targeted Therapy: A Meta-Analysis

**DOI:** 10.3390/diagnostics11030475

**Published:** 2021-03-08

**Authors:** Kota Yokoyama, Junichi Tsuchiya, Yuji Nakamoto, Ukihide Tateishi

**Affiliations:** 1Department of Diagnostic Radiology, Tokyo Medical and Dental University, Tokyo 113-8510, Japan; tuwu11@gmail.com (J.T.); ttisdrnm@tmd.ac.jp (U.T.); 2Department of Diagnostic Imaging and Nuclear Medicine, Kyoto University, Kyoto 606-8507, Japan; ynakamo1@kuhp.kyoto-u.ac.jp

**Keywords:** [^18^F]FDG PET, PET/CT, GIST, molecular targeted therapy, treatment response assessment

## Abstract

To assess the additional value of 2-deoxy-2-[^18^F] fluoro-d-glucose ([^18^F]FDG) positron emission tomography (PET) or PET/CT over conventional morphological imaging techniques in the treatment response assessment of gastrointestinal stromal tumor (GIST) to molecular targeted therapy (MTT), we performed a meta-analysis of all the available studies to compare the predictive value of [^18^F]FDG PET or PET/CT and conventional imaging techniques for assessing the response to MTT in GIST. We determined the sensitivities and specificities across studies, we calculated the positive and negative likelihood ratios (LR) and made summary receiver operating characteristic curves (SROC) using hierarchical regression models. Pooled analysis included 4 studies comprising 88 patients. The performance characteristics in [^18^F]FDG PET or PET/CT and CT were as follows: sensitivity, 89% (95% confidence interval (CI) 78, 95), 52% (39, 64); specificity, 65% (44, 83), 92% (75, 99); diagnostic odds ratios (DOR), 5.8 (2.0, 16.8 4.9 (1.5, 16.1); positive LR, 1.9 (1.1, 3.4), 3.0 (1.1, 8.1); and negative LR, 0.23 (0.03, 1.6), 0.66 (0.42, 1.0), respectively. In SROC curves, the area under the curve (AUC) was 0.81 (SE, 0.11) and 0.71 (SE, 0.13) and the Q* index was 0.74 and 0.66, respectively. [^18^F]FDG PET/CT had higher sensitivity, while DOR and SROC curves showed better diagnostic performance in [^18^F]FDG PET and PET/CT studies as compared to CT.

## 1. Introduction

Gastrointestinal stromal tumor (GIST), the most common mesenchymal tumor of the gastrointestinal (GI) tract, has an annual incidence of 6.8–14.5 per million individuals [1,2,3]. GISTs are often considered resistant to chemotherapy and are insensitive to irradiation; further, the lack of effective treatments and its metastatic nature generally results in a poor prognosis [4,5,6,7]. GIST usually develops from oncogenic mutations in the tyrosine kinase receptor KIT (CD117) [8], and/or platelet-derived growth factor receptor-α (PDGFR-α) [9]; the identification of c-KIT and PDGFR-α has resulted in the establishment of new therapeutic approaches based on therapies targeting the receptors, namely, MTT. After the introduction of tyrosine kinase inhibitors (TKIs), such as imatinib methylate in 2001 for GIST treatment, the prognosis and therapeutic outcome of this tumor entity have considerably improved. During the previous decade, the use of multiple TKIs other than imatinib has led to an increase in the GIST median survival to nearly 5 y as compared to the previous average survival duration of 9–20 months [10,11]. The introduction of molecular targeted therapy (MTT) in GIST has improved the prognosis and therapeutic outcome; however, the expensiveness of the treatment has increased the importance of appropriate diagnostic tools, encouraging early assessment and confirmation of a therapeutic response. Conventional morphological criteria based on changes in the tumor size, such as RECIST, were not sufficient to assess the treatment effect and appear to underestimate the early response to MTT [12]. Current evidence shows that the 2-deoxy-2-[^18^F] fluoro-d-glucose positron emission tomography ([^18^F]FDG PET) is a sensitive tool for the evaluation of early therapeutic response to MTT in GIST [13,14] [^18^F]FDG PET may also enable us to detect TKI resistance [14,15]. Many studies have shown the value of [^18^F]FDG PET in MTT response assessment in GIST; however, the conventional morphological criteria with CT remains the gold standard, with its use being limited in many countries [16]. Therefore, strong evidence is required in favor of the routine use of [^18^F]FDG PET in clinical decision-making. The present study was designed to perform a meta-analysis of all the available studies and assess the advantage of additional [^18^F]FDG PET or PET/CT over CT or magnetic resonance imaging (MRI) in the assessment of the treatment response to MTT in GIST.

## 2. Materials and Methods

### 2.1. Data Sources Eligibility

We searched Medline (from 2000 to October 2020), SCOPUS, and Biological Abstracts. We used a search algorithm that was based on a combination of the following terms: (1) [^18^F]FDG, [^18^F]FDG PET, or [^18^F]FDG PET/CT, (2) gastrointestinal stromal tumor, and GIST (3) molecular targeted therapy, TKIs, imatinib, sunitinib, and regorafenib. We did not apply any language restrictions. The two reviewers independently assessed the potentially relevant citations for inclusion, and disagreements were resolved via consensus. Referenced articles of the retrieved studies were screened to identify additional studies.

Studies were included if they fulfilled the following inclusion criteria: (a) Study included GIST patients treated with MTT, such as imatinib and sunitinib. (b) [^18^F]FDG PET or PET/CT in addition to CT or MRI was used to assess the treatment response. (c) Data of treatment outcome (i.e., complete remission (CR), partial remission (PR), stable disease (SD), progressive disease (PD), and nonassessable patients (NA)) in [^18^F]FDG PET or PET/CT and CT or MRI could be extracted independently. (d) Histopathologic confirmation or imaging follow-up was considered as the reference standard. (e) When data were presented in more than one article, the article with most details or the most recent article was chosen. Studies were excluded if data were unavailable for deriving 2 × 2 tables to draw summary receiver operating characteristic curve (SROC). Reviews, letters, case reports, and meeting abstracts were also excluded.

### 2.2. Data Extraction

We extracted the data from eligible studies independently and resolved any issues by consensus. We recorded the author names, journal, publication year, origin country, number of patients, age, inclusion and exclusion criteria, study design, treatment details, imaging details, namely imaging system (PET or PET/CT), number of experts interpreting the images, and definition of positive test result (qualitative or quantitative). The numbers of CR, PR, SD, PD, and NA for each modality were also recorded. In addition, we extracted the numbers of patients who had treatment effect as good responders (CR + PR) and poor responders (SD + PD) of [^18^F]FDG PET and CT, and 2 × 2 tables were created, including the numbers of TP, FP, FN, and TN. Data extracted from publications alone were deemed adequate for this meta-analysis without contacting the authors for more information. Two reviewers (KY and JT) individually used the Quality Assessment of Diagnostic Accuracy Studies-2 (QUADAS-2) tool that is widely used to assess the quality of systematic reviews of diagnostic studies [17].

### 2.3. Statistical Analysis

To evaluate the diagnostic value of [^18^F]FDG PET or PET/CT in MTT response assessment of GIST, we calculated the pooled estimates of sensitivity, specificity, positive likelihood ratio (PLR), negative likelihood ratio (NLR), diagnostic odds ratios (DORs), and their 95% CI both in CT and [^18^F]FDG PET or PET/CT. The likelihood ratios combine the sensitivity and specificity in their calculation. The PLR is defined as the ratio of sensitivity over (1-specificity), and the NLR is defined as the ratio of (1-sensitivity) over specificity. The DOR was provided by the ratio of PLR relative to NLR, with higher values indicating better performance. In addition, SROC curves were drawn, with the area under the curve (AUC) and Q* index obtained. Q* index is the best statistical method to reflect the diagnostic value, it is defined by the point where sensitivity and specificity are equal, representing the point closest to the ideal top-left corner of the SROC space [18]. The degree of heterogeneity among different studies was tested using the I2 test. When significant heterogeneity was observed, namely the I2 value was >50%, a random-effect model was applied; in other cases, a fixed-effect model was used [19]. Analyses were performed using Meta-Disc v. 1.4 (XI Cochrane Colloquium, Barcelona, Spain) and RevMan 5.3.

## 3. Results

### 3.1. Article Search

Our literature search yielded 122 articles; 98 nonrelevant articles were excluded upfront after reading the abstract. To assess the eligibility of the remaining 24 articles, we retrieved the corresponding full texts. Of these 24 studies, data on patients for deriving 2 × 2 tables were available in four for both [^18^F]FDG PET or PET/CT, and CT [12,20,21,22]. A graphical summary of the article selection process is provided in Figure 1. The quality assessment of the included four studies is shown in Figure 2. A total of 91 patients were analyzed for the diagnostic accuracy to assess the treatment effect of MTT. All 91 patients were evaluated by CT, of which 88 were also evaluated by [^18^F]FDG PET or PET/CT.

Among the four eligible studies, three studies were prospective, and 1 was a retrospective study. All the studies enrolled patients with GIST treated with MTT. Two studies used PET, while one used PET/CT, and one used coincidence PET camera. The amount of radiotracer was 250 Mbq, 6 Mbq/kg, 7.5 Mbq/kg, 10–15 mCi, respectively and the time interval was 60 min in all the studies. The evaluation of study results was performed qualitatively in all studies. Reference standard consisted of CT, [^18^F]FDG PET or PET/CT, and clinical follow-up. The detailed characteristics of included studies are summarized in Table 1.

### 3.2. Pooled Diagnostic Performance (Meta-Analysis)

For CT and [^18^F]FDG PET or PET/CT, we combined the SROC curves, and their corresponding findings, in Figure 3 and Figure 4, respectively. Per imaging modality, the findings are as described below.

CT

This pooled analysis included four studies comprising 91 patients. The performance characteristics were sensitivity, 52% (39, 64); specificity, 92% (75, 99); DOR, 4.9 (1.5, 16.1); positive LR, 3.0 (1.1, 8.1); and negative LR, 0.66 (0.42, 1.0). In SROC curves, the AUC was 0.71 (SE, 0.13), and the Q* index was 0.66.

[^18^F]FDG PET/CT or PET/CT

This pooled analysis included four studies comprising 88 patients. The performance characteristics were sensitivity, 89% (95% CI 78, 95); specificity, 65% (44, 83); DOR, 5.8 (2.0, 16.8); positive LR, 1.9 (1.1, 3.4); and negative LR, 0.23 (0.03, 1.6). In SROC curves, the AUC was 0.81 (SE, 0.11) and the Q* index was 0.74.

Comparison of Imaging Modalities

In sum, [^18^F]FDG PET or PET/CT had the higher sensitivity (89%) and DOR (5.8) and the SROC curves showed excellent diagnostic performance of [^18^F]FDG PET or PET/CT studies (Figure 3 and Figure 4).

## 4. Discussion

When compared with those of CT, the diagnostic performance of [^18^F]FDG PET or PET/CT was excellent in assessing the treatment response to MTT in GIST.

Overall, the value of [^18^F]FDG PET was higher for detection of early treatment response (higher sensitivity) than CT. Previous studies of [^18^F]FDG PET/CT for other before and after induction chemotherapy have shown a significant association between early metabolic response and histopathologic tumor regression. The value of [^18^F]FDG PET for predicting treatment response in GIST patients treated with imatinib or other systemic therapy is well established. However, it is unclear whether additional [^18^F]FDG PET or PET/CT was useful for patients who had already been evaluated with CT or MRI. Although one meta-analysis assessed the value of [^18^F]FDG PET in the prediction of therapeutic response of GIST patients [14], to our knowledge, this is the first meta-analysis that compared the diagnostic ability of [^18^F]FDG PET or PET/CT with CT in assessing the treatment response to MTT in GIST.

Based on the results of this meta-analysis, the performance of [^18^F]FDG PET or PET/CT in the assessment of the treatment effect is satisfactory because high percentage of true-positive and a low percentage of false-negative results were found. In contrast, the specificity in [^18^F]FDG PET or PET/CT was lower than that in CT. The possible reason for this is a higher detection rate of transient partial remission, reflecting the high accuracy of [^18^F]FDG PET or PET/CT, even in the advanced stage patients who consequently had poor prognosis that probably led to an increasing rate of false-positive results when the reference standard was based on the patient follow-up [20].The fact that most false-positive cases were in PR and very few CR patients showed bad prognosis might support the idea. Therefore, advanced stage GIST patients showing early PR in FDG PET with poor response in CT should be carefully followed up. Although the number of studies included in this study is small, the results that [^18^F]FDG PET/CT is superior to CT in the response evaluation of the target drug of GIST were reasonable. Moreover, there is no significant differences between the studies, and we believe it is sufficient to conclude without further study accumulation.

The present study has certain limitations. First, the diagnostic value of [^18^F]FDG PET in GIST patient was reported without adjusting for potential confounders, such as grading and staging of patients. Moreover, the source of heterogeneity (publication bias) that results in systematic differences in effect size estimates derived from small versus large studies was included. The available data are consistent with great improvement in sensitivity with [^18^F]FDG PET or PET/CT over CT in the studies enrolled. Although the confidence intervals include a low sensitivity for CT and a low specificity for [^18^F]FDG PET, the number studies were limited because only those studies with data available for deriving 2 × 2 tables to draw SROC curve were included. There was a risk of subjective interpretation because the interpretation of [^18^F]FDG PET or PET/CT was performed by two reviewers qualitatively in all the studies; however, the presence/absence of blinding was unclear. In the definition of good response or poor response, most studies lacked not only histologic confirmation, but also other diagnostic biomarker such as MRI, tumor markers, or other blood test data. However, biopsies of multiple lesions in one patient were not substantial, and whole-body MRI was limited to clinical use in the entire study period. Another limitation of this study was that there were few articles that used the latest diagnostic devices, i.e., there was only one PET/CT study, and there were no PET/MRI or whole-body MRI studies. Further validation studies using [^18^F]FDG PET or PET/CT compared with PET/MRI or whole-body MRI are warranted. Although the cost and radiation exposure of each procedure are important concerns, not only for the patient, but also for the health care system, we were unable to analyze them because of the limited number of published articles. Selecting the most appropriate imaging test in any clinical situation depends upon the circumstances of the patient, the expertise, and equipment available at the treating site and procedure costs. The goal is certainly to use the most cost-effective imaging method to allow accurate diagnosis and prompt treatment of this common, complex, and costly problem. This will require further prospective studies on a larger sample size with a direct comparison between the different radiological and nuclear medicine techniques.

Thus, our study provides evidence to support the possible application of additional [^18^F]FDG PET or PET/CT for assessing the treatment response to MTT in GIST patients who have already been assessed using CT. [^18^F]FDG PET has higher sensitivity for detection of early treatment response than CT; however, the false-positive rate may increase in advanced GIST patients. Thus, patients should be carefully followed up using other diagnostic biomarkers as well. Further research is required to determine whether there is indeed an incremental diagnostic improvement with [^18^F]FDG PET or PET/CT over other imaging methods with consideration of radiation dose, cost-effectiveness, and potential complications against the yield of information.

## Figures and Tables

**Figure 1 diagnostics-11-00475-f001:**
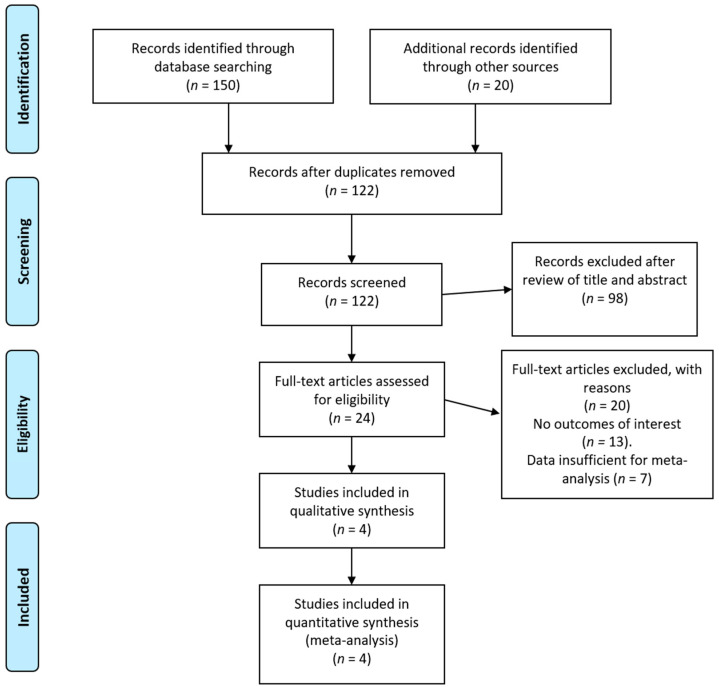
PRISMA flow diagram of included articles. PRISMA, Preferred Reporting Items for Systematic Reviews and Meta-Analyses.

**Figure 2 diagnostics-11-00475-f002:**
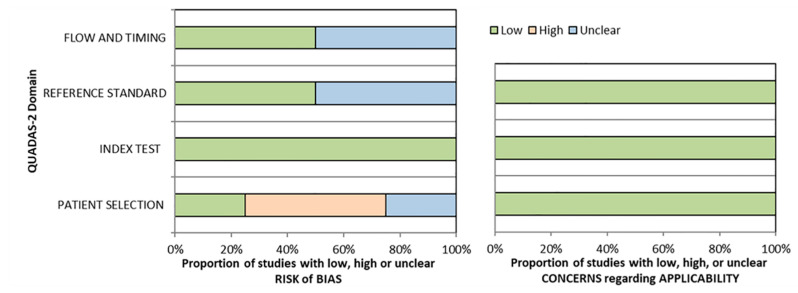
Quality assessment of the included studies using Quality Assessment of Diagnostic Accuracy Studies 2 (QUADAS-2).

**Figure 3 diagnostics-11-00475-f003:**
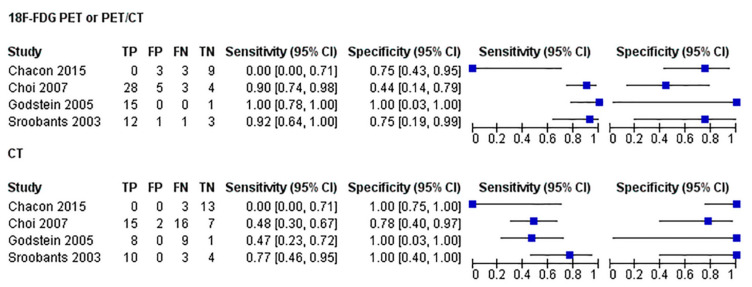
Forest plots (of sensitivity and specificity) of [^18^F]FDG PET or PET/CT and CT and pooled the diagnostic performance of the imaging techniques. For each test, we combined the SROC curves and their corresponding findings.

**Figure 4 diagnostics-11-00475-f004:**
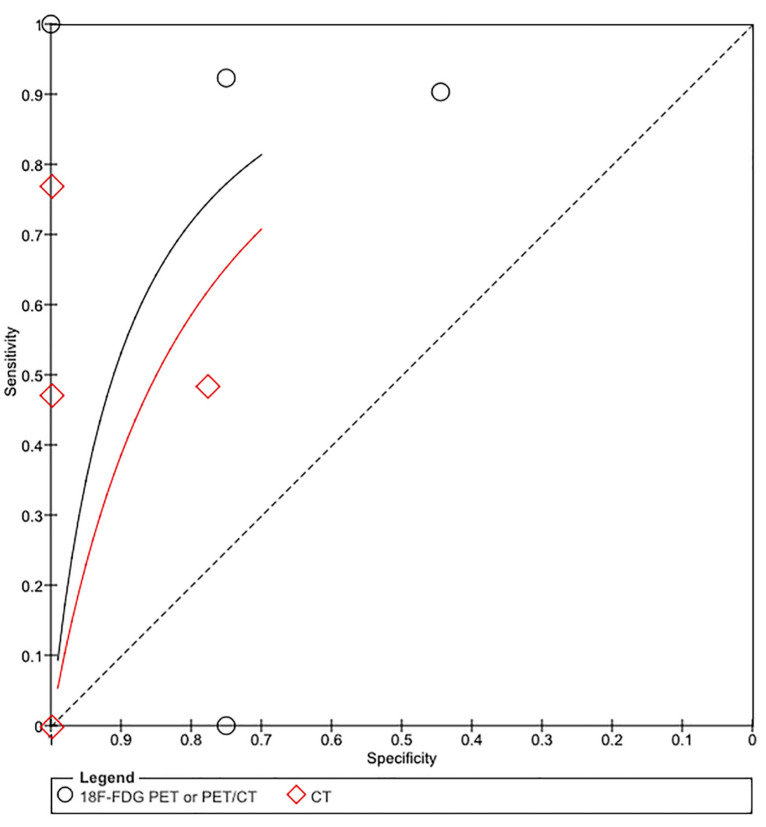
Summary receiver operating characteristic (SROC) curve for the diagnostic performance of [^18^F]FDG PET or PET/CT (black curve) and CT (red curve). The size of the circle diamond indicates the weight of each study in [^18^F]FDG PET or PET/CT and CT, respectively. The area under the SROC curve is 0.81 for [^18^F]FDG PET or PET/CT and 0.76 for CT.

**Table 1 diagnostics-11-00475-t001:** Study characteristics for selected studies.

Author	Year	Ref.	Country	No. of Patients	Mean Age (Range)	Design	Treatment Protocol	Modality	FDG Dose	Time Interval	Fastig Time	Diagnosis	No. of Assessors	Reference Standard	Duration
Sroobants et al.	2003	[21]	Belgium	17	55 (38–70)	P	Imatinib 400–800 mg/day	PET	6 MBq/kg	60 min	6 h	Qualitive	NR	FU	NR
Godstein et al.	2005	[22]	Australia	18	NR	P	Imatinib 400-800 mg/day	co-PET	250 MBq	60 min	6 h	Qualitive	NR	FU	2001–2003
Choi et al.	2007	[12]	USA	40	51 (13–76)	R	Imatinib 400–1000 mg/day	PET	10–15 mCi	60 min	6 h	Qualitive	NR	FU	2000.12–2001.9
Chacón et al.	2015	[20]	Argentina	16	49 (25–73)	P	Imatinib 400–800 mg/day	PET/CT	7.5 MBq/kg	60 min	6 h	Qualitive	2	FU	2006.3–2008.7

NR: not reported, P: prospective, R: retrospective, FU: follow up.

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
