# Peer review of "Additional Value of [18F]FDG PET or PET/CT for Response Assessment of Patients with Gastrointestinal Stromal Tumor Undergoing Molecular Targeted Therapy: A Meta-Analysis"

_diagnostics, 2021, doi:10.3390/diagnostics11030475_

Round 1
Reviewer 1 Report
Very nice topic and interesting results.
Author Response
We would like to thank you for taking the time to review this study and for your understanding. We hope that this revision also persuades you to accept our submission.
Reviewer 2 Report
The manuscript is quite well written and organized.
Author Response
We appreciate your kind consideration of our paper. We hope that this revision also persuades you to accept our submission.
Reviewer 3 Report
In this meta-analysis on the additional value of [18F]FDG PET or PET/CT for response assessment of patients with GIST undergoing molecular targeted therapy few articles and patients are included. Therefore this paper fails to provide significant results.
The authors made great efforts but the issue is the limited literature data available. This issue cannot be solved by a revision of the article, but more literature data are needed.
Furthermore, I would suggest to focus the analysis on PET/CT only because, to date, this is the standard PET method.
Author Response
General Comments
We thank the reviewer for the helpful comments and suggestions, which we believe have strengthened the manuscript substantially. We have addressed all comments in our revised manuscript and discuss them below on a point-by-point basis. All changes to the revised manuscript are presented in red font in the manuscript.
Comment 1. In this meta-analysis on the additional value of [18F]FDG PET or PET/CT for response assessment of patients with GIST undergoing molecular targeted therapy few articles and patients are included. Therefore this paper fails to provide significant results. The authors made great efforts but the issue is the limited literature data available. This issue cannot be solved by a revision of the article, but more literature data are needed.
Response:
We would like to thank you for taking the time to review this study and for your valuable advice. It is true that the number of included studies in the present study is small, however the result that the PET/CT is better than CT in response assessment of targeted agents for GIST is reasonable, and there was no significant differences between the studies. Therefore, we judged that the result was credible without further case accumulation. New drugs, including molecular-targeted drugs, are expensive, and whether PET/CT evaluation is actually useful in terms of health economics and affects patient outcomes should be an issue for the future.
In response to your suggestions, we have added the following to the discussion.
Line 186
Although the number of studies included in this study is small, the results that [18F]FDG PET/CT is superior to CT in the response evaluation of the target drug of GIST were reasonable. Moreover, there is no significant differences between the studies, and we believe it is sufficient to conclude without further study accumulation.
Comment 2. Furthermore, I would suggest to focus the analysis on PET/CT only because, to date, this is the standard PET method.
Response:
Thank you for your suggestion. Currently, the standard PET method is PET/CT as you pointed out, but the usefulness of [18F]FDG PET/CT in detecting early therapeutic responses is mainly due to metabolic evaluation by PET rather than morphological evaluation by CT. In addition, the usefulness of [18F]FDG PET or PET/CT for therapeutic response assessment is discussed in other tumors as well. From that point, we would like to emphasize the usefulness of metabolic evaluation by PET or PET/CT, and we think that we do not necessarily have to limit it to PET/CT studies.
Round 2
Reviewer 3 Report
As stated in my previous report, unfortunately the issue of this meta-analysis remains the limited literature data available. This issue cannot be solved by a revision of the article, but more literature data are needed to provide a reasonable conclusion as proposed by the authors.
It is well known that the number of included studies (and patients) influences the generalizability the conclusions derived by the analysis cannot be generalized.
It could be more useful for the readers, clinicians and researhcers to wait for more literature data suppoting the conclusions provided by the authors.